

# A guide for deploying deep learning in LHC searches: How to achieve optimality and account for uncertainty

**Benjamin Nachman**[⋆]

Physics Division, Lawrence Berkeley National Laboratory, Berkeley, CA 94720, USA

⋆ bpnachman@lbl.gov

## Abstract

Deep learning tools can incorporate all of the available information into a search for new particles, thus making the best use of the available data. This paper reviews how to optimally integrate information with deep learning and explicitly describes the corresponding sources of uncertainty. Simple illustrative examples show how these concepts can be applied in practice.

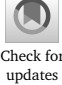

# 1 Introduction

Since the first studies of deep learning[1] in high energy physics (HEP) [3,4], there has been a rapid growth in the adaptation and development of techniques for all areas of data analysis (see Ref. [5–7] and references therein). One of the most exciting prospects of deep learning is the opportunity to exploit all of the available information to significantly increase the power of Standard Model measurements and searches for new particles.

While the first analysis-specific deep learning results are only starting to become public (see e.g. [8–10]), analysis-non-specific deep learning has been used for a few years starting with flavor tagging [11,12]. In addition, there are a plethora of experimental[2] and phenomenological [3] studies for additional methods which will likely be realized as part of physics analyses in the near future.

The goal of this paper is to clearly and concisely describe how to achieve optimality and account for uncertainty using deep learning in LHC data analysis. One of the most common questions when an analysis wants to use deep learning is '...but what about the uncertainty on the network?' Ideally the discussion here will help clarify and direct this question. The concepts of accuracy/bias and optimality/precision will be distinguished when covering uncertainties. In particular, most applications of neural networks do not result in uncertainties on the networks themselves related to the accuracy of the analysis. This statement will be qualified in detail. The exposition is based on a mixture of old and new insights, with references given to the foundational papers for further reading – although it is likely that even older references exist in the statistics literature. Section 2 introduces a simple example that will be used for illustration throughout the paper. Sections 3 and 4 discuss achieving optimality and including uncertainties, respectively. A brief discussion with future outlook is contained in Sec. 5.

# 2 Illustrative Model

One of the most widely used techniques to search for new particles in HEP is to seek out a localized feature on top of a smooth background from known phenomena, also known as the 'bump hunt'. This methodology was used to discover the Higgs boson [13,14] and has a rich history, dating back at least to the discovery of the $\rho$ meson [15]. The localized feature in these searches is often the invariant mass of two or more decay products of the hypothetical new resonance. A simplified version of this search is used as an illustrative example.

Consider a simple approximation to the bump hunt where the background distribution is uniform and the signal is a $\delta$-function. Let $X$ be a random variable corresponding to the bump

---

[1]Here, 'deep learning' means machine learning with modern neural networks. These networks are deeper and more flexible than artificial neural networks from the past and can now readily process high-dimensional feature spaces with complex structure. In the context of classification, deep neural networks are powerful high-dimensional likelihood-ratio approximators, as described in later sections. Machine learning has a long history in HEP and there are too many references to cite here – see for instance Ref. [1] and the many papers that cite it and came before going back to at least Ref. [2].

[2]These are mostly contained in ATLAS/CMS/LHCb/ALICE performance notes and theses - see e.g. studies for missing energy [ATL-PHYS-PUB-2019-028], (non $b$-jet) jet tagging [CMS-DP-2017-027, CMS-DP-2017-049,ATL-PHYS-PUB-2018-014,ATL-PHYS-PUB-2017-017,ATL-PHYS-PUB-2017-004], simulation [ATL-SOFT-PUB-2018-001], calibration [ATL-PHYS-PUB-2018-013], track reconstruction [ATL-PHYS-PUB-2018-002], Data quality [74], and much more.

[3]See for instance Ref. [5–7] for recent reviews and e.g. Ref. [34,46–55,75–81] for anomaly detection, Ref. [31,82–86,86–96,96–110] for generative models, Ref. [45,67–69,72,73,111–113] for decorrelation techniques, Ref. [24,28–31,33,41–43,66,100,114–119] for inference, Ref. [3,4,40,60,109,120–164] for tagging, and various other applications in Ref. [165–167].

hunt feature, which can be thought of as the invariant mass of two objects such as high energy jets. Under this model, $X|\text{background} \sim \text{Uniform}(-0.5, 0.5)$ and $X|\text{signal} = \delta(0)$. Following the analogy with jets, the random variable $X$ will be built from two other random variables representing the jet energies. As the energy of the daughter objects would be approximately half of the mass of the parent, let the decay products be $X_0, X_1 = X/2$. Detector distortions will perturb the $X_i$. Let $Y_i$ represent the measured versions of the $X_i$. The experimental resolution will affect $X_0$ and $X_1$ independently. To model these distortions, let $Y_i = X_i + Z_i$ where $Z_i \sim \mathcal{N}(0, \sigma_i^2)$. Figure 1 schematically illustrates the connection between the back-to-back decay of a massive particle produced at rest and the approximation used here.

For simplicity, all arithmetic is performed 'mod $[-0.5, 0.5]$' so that everything can be visualized inside a compact interval. This means that an integer is added or subtracted until the resulting value is in the range $[-0.5, 0.5]$. For example, if $X_0 = 0.4$ and $Z_0 = 0.2$ then $Y_0 = -0.4$. The distributions of $Y_0 \pm Y_1$ are shown in Fig. 2.

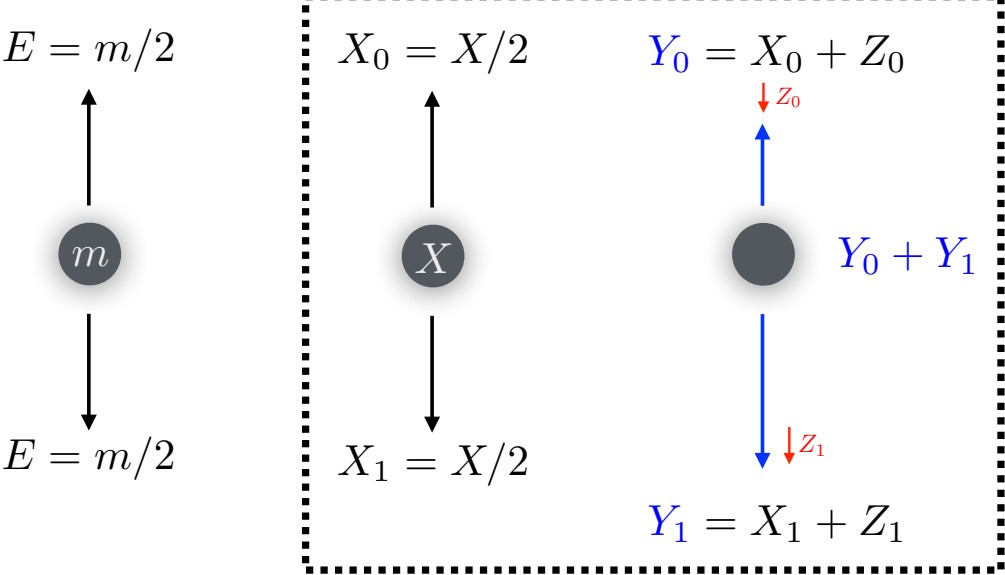

Figure 1: A schematic diagram of the illustrative model. An actual two-body decay in the parent particle rest frame is shown on the left. In the simple example, the two energies are collinear and detector effects ($Z_i$) only modify the magnitude but not the direction.

The reconstructed mass is given by $Y_0 + Y_1$ and can be used for the resonance search. By construction, $Y_0 - Y_1$ contains no useful information for distinguishing the signal and background processes (see right plot of Fig. 2), but is sensitive to the value of $\sigma$. Two scenarios for $\sigma_i$ will be investigated in later sections: (1) $\sigma_i = \sigma$ is constant and the same for all events[4] and (2) $\sigma_i$ is different, but known for each event. The former is the usual case for a global systematic uncertainty and in this simple example is analogous to the jet energy scale resolution [16,17]. The latter is the opposite extreme, where there is a precise event-by-event constraint. The physics analog of (2) would be jet-by-jet estimates of the jet energy uncertainty or the number of additional proton-proton collisions ('pileup') in an event, which degrades resolutions, but can be measured for each bunch crossing.

The experimental goal is to identify if the data are consistent with the background only, or if there is evidence for a non-zero contribution of signal.

---

[4]In practice, $\sigma$ may vary from event-to-event in a known way, but there is one global nuisance parameter.

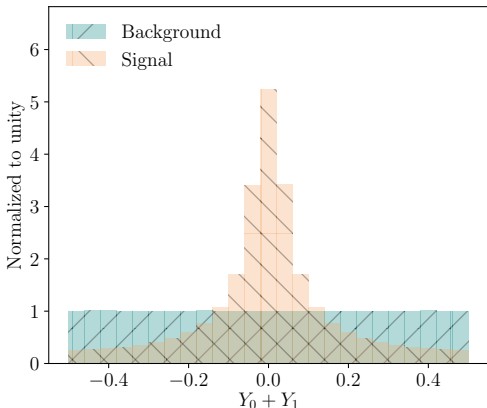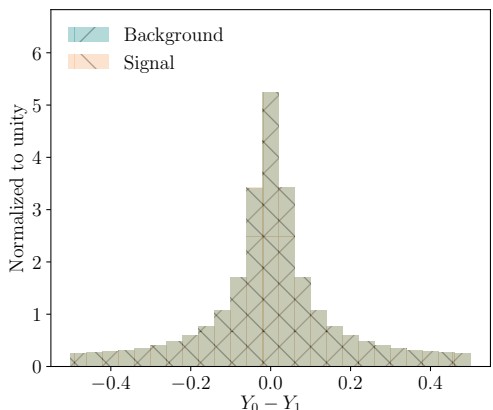

Figure 2: A histogram of $Y_0 + Y_1$ (left) and $Y_0 - Y_1$ (right). As the true signal is a $\delta$-function and resolution effects are independent for both 'decay products', the measured distribution of $Y_0 + Y_1$ for the signal is the same as $Y_0 - Y_1$ for both signal and background. A constant value of $\sigma$ is used for all events.

## 3 Achieving Optimality

### 3.1 Overview

The usual way of performing an analysis with $Y = (Y_0, Y_1)$ is to define a *signal region* and then count the number of events in data and compare it to the number of predicted events[5]. For a fixed and known value[6] of $\sigma$, and a particular signal model, the probability distribution for the number of observed events follows a Poisson distribution:

$$p(|\mathbf{Y}||\mu) = \frac{(\mu S + B)^{|\mathbf{Y}|}e^{-(\mu S + B)}}{|\mathbf{Y}|!}, \tag{3.1}$$

where $\mathbf{Y} = \{Y_i\}$, and $S, B$ are the predicted signal and background event yields. Viewed as a function of $\mu$ for fixed $|\mathbf{Y}|$, Eq. (3.1) is the likelihood $\mathcal{L}(\mu)$. One can express $S = \int \epsilon \, \sigma_{\text{physics}} \, L \, dt$, where $\epsilon$ is an event-selection efficiency, $\sigma_{\text{physics}}$ is the cross-section for producing events, and $L$ is the instantaneous luminosity. These yields may be derived directly from simulation or completely / partially constrained from *control regions* in data. The parameter $\mu$ distinguishes the null hypothesis[7] $\mu = 1$ from the alternative hypothesis $\mu = 0$.

By the Neyman-Pearson lemma [19], the most *powerful*[8] test for this analysis is performed with the likelihood ratio test statistic:

$$\lambda_{\text{LR}}(|\mathbf{Y}|) = p(|\mathbf{Y}||1)/p(|\mathbf{Y}||0). \tag{3.2}$$

At the LHC, it is more common to use the profile likelihood ratio instead:

$$\lambda_{\text{PLR}}(\mu) = p(|\mathbf{Y}||\mu)/\max_{\mu} p(|\mathbf{Y}||\mu). \tag{3.3}$$

---

[5]See Ref. [18] for a review of statistical procedures in HEP.

[6]Section 4 will revisit the above setup when $\sigma$ is not known.

[7]This is for setting limits. For discovery, the null hypothesis is the Standard Model, $\mu = 0$.

[8]This means that for a fixed probability of rejecting the null hypothesis when it is true, this test has the highest probability of rejecting the null hypothesis when the alternative is true.

There is no uniformly most powerful test in the presence of nuisance parameter, but Eq. (3.3) is still the most widely-used test statistic. Two related quantities are the $CL_{S+B}$ and $CL_B$, which are the $p$-values associated with the Eq. (3.3) under the null and alternative hypotheses:

$$CL_{S+B} = \Pr\big(\lambda_{PLR}(1) > \lambda_{PLR}^{data}(1)|\mu = 1\big) \tag{3.4}$$

$$CL_B = \Pr\big(\lambda_{PLR}(1) > \lambda_{PLR}^{data}(1)|\mu = 0\big), \tag{3.5}$$

where $\lambda_{PLR}^{data}$ is the observed test statistics. In the absence of signal, $\underset{\mu}{\arg\max}\, p(|\mathbf{Y}||\mu) \approx 0$, so this is nearly the same as $\lambda_{LR}(|\mathbf{Y}|)$. A generic feature of hypothesis tests where the two hypothesis do not span the space of possibilities is that both the null and alternative hypothesis can be inconsistent with the data. The HEP solution to this phenomenon is to exclude a model if the value of $CL_S = CL_{S+B}/CL_B$ is small instead of $CL_{S+B}$ [20, 21]. The $CL_S$ does not maximize statistical power and is not even a proper $p$-value. Other proposals for regulating the $p$-value have been discussed (see e.g. Ref. [22] or Sec. 7.1. in Ref. [23]) but are not used in practice. The results that follow will focus on $\lambda_{LR}$ and the lessons learned are likely approximately valid for the HEP solution as well[9].

The usual way deep learning is used in analysis is to train a classifier $f(Y) : \mathbb{R}^2 \to \mathbb{R}$ and count the number of events with $f(Y) > c$, where $c$ is chosen to maximize significance. These count data are then analyzed using the same likelihood ratio approach described above. Some analyses use $f(Y)$ to make categories for performing a multi-dimensional statistical analysis. Section 3.2 discusses the differences between threshold cuts and binning and describes how to do an unbinned version that can use all of the available information.

## 3.2 Cuts, bins, and beyond

Suppose momentarily that the probability density for $Y$ is piece-wise constant over a finite number of patches $P$. Each patch is independent, so the full likelihood is the product over patches:

$$p(\mathbf{Y}|\mu) = \prod_{i \in P} \Pr(\mathbf{Y} \text{ in patch } i|\mu) = \frac{(\mu S_i + B_i)^{\sum_{j=1}^{n} \mathbb{I}[Y_j \in i]} e^{-(\mu S_i + B_i)}}{\left(\sum_{j=1}^{n} \mathbb{I}[Y_j \in i]\right)!}, \tag{3.6}$$

where $\mathbb{I}[\cdot]$ is an indicator function that is one when $\cdot$ is true and zero otherwise. This is product over terms like Eq. (3.1), one for each patch. Defining $n_i = \sum_{j=1}^{n} \mathbb{I}[Y_j \in i]$, one can rewrite Eq. (3.6) as

$$p(\mathbf{Y}|\mu) = e^{-(\mu S+B)} \prod_{i \in P} \frac{(\mu S_i + B_i)^{n_i}}{(\mu S + B)^{n_i}} \frac{(\mu S + B)^{n_i}}{n_i!} \propto \frac{(\mu S + B)^{|\mathbf{Y}|} e^{-(\mu S+B)}}{|\mathbf{Y}|!} \prod_{j=1}^{|\mathbf{Y}|} p_{\mu S+B}(Y_j), \tag{3.7}$$

where $p_{\mu S+B}(Y)$ is the probability to observe $Y$. Equation (3.7) is often called the *extended likelihood* [26]. The important feature about Eq. (3.7) is that it does not depend on the patches and so in the continuum limit, it is also appropriate for the full phase space likelihood where $p_{\mu S+B}$ is now a probability density.

The likelihood ratio statistic using Eq. (3.7) is then given by

---

[9]The neural network approximates below naturally extend to the profiling case via parameterized neural networks [24, 25], where the dependence on $\mu$ is explicitly learned.

$$\lambda_{\text{full LR}}(\mathbf{Y}) = \lambda_{\text{LR}}(|\mathbf{Y}|) \times \prod_{i=1}^{|\mathbf{Y}|} \frac{p_{S+B}(Y_i)}{p_B(Y_i)}. \tag{3.8}$$

The optimal use of the full phase space would then be to exclude the model if $\lambda_{\text{full LR}}(\mathbf{Y})$ is small. Especially when $Y$ is high-dimensional, $p_{S+B}(Y)$, $p_B(Y)$, and $\lambda(Y) = p_{S+B}(Y)/p_B(Y)$ are not known analytically. Using a small number of bins is an approximation to Eq. (3.8) and can provide additional power beyond Eq. (3.1). Adding bins is only helpful if the events in each bin have a different likelihood ratio. The loss functions used in deep learning ensure that the network output is monotonically related to the likelihood ratio (more on this below). Therefore, bins chosen based on the output of a neural network typically enhance the statistical power of an analysis. Unless the likelihood ratio only takes on a small number of values, binning will necessarily be less powerful than an unbinned approach using the full likelihood for $Y$. Using the output of a neural network to construct bins does help to reduce the potentially high-dimensional problem to a one-dimensional one. However, a small number of bins may not be sufficient to capture all of the salient structures in the likelihood ratio, especially when $Y$ is high-dimensional.

A powerful way of estimating the second term in Eq. (3.8) is to use deep learning – instead of or in addition to placing a threshold cut[10]. For making bins, it is sufficient for the neural network output to be monotonic with the likelihood ratio. However, Eq. (3.8) requires that the network output be proportional to the likelihood ratio. This needs some care when training the neural network and interpreting its output. For example, suppose that a neural network $\text{NN} : \mathbb{R}^2 \to [0, 1]$ is trained with the standard cross-entropy loss function:

$$\text{Loss(NN)} = -\sum_{i \in S+B} \log(\text{NN}(Y_i)) - \sum_{i \in B} \log(1 - \text{NN}(Y_i)), \tag{3.9}$$

where the output range $[0, 1]$ can be achieved with a non-linear function in the last layer of the neural network that outputs a number between 0 and 1 such as the commonly used sigmoid. Appendix A shows that such a neural network will asymptotically (more on this later) learn $p(S + B|Y)$. This is not the likelihood ratio, but some symbolic manipulation shows that it is monotonically related to it:

$$\begin{aligned} p(S + B|Y) &= \frac{p(Y|S + B)p(S + B)}{p(Y)} \\ &= \frac{p(Y|S + B)p(S + B)}{p(Y|S + B)p(S + B) + p(Y|B)p(B)} \\ &= \frac{\lambda(Y)}{p(B)/p(S + B) + \lambda(Y)}, \end{aligned} \tag{3.10}$$

where $p$ denotes a probability or probability density. If instead of directly using the NN output, one uses $\tilde{\lambda}(Y) = \text{NN}(Y)/(1 - \text{NN}(Y))$, then a similar calculation to Eq. (3.10) shows that $\tilde{\lambda}(Y) \propto \lambda(Y)$, where the proportionality constant is the ratio of the background and signal dataset sizes used during the NN training. The modified function $\tilde{\lambda}(Y)$ would be appropriate as a surrogate to the second term in Eq. (3.8). Interestingly, the same $\tilde{\lambda}(Y)$ works when the mean squared error loss is used. One can even choose a non-standard loss that directly learns a function proportional to the likelihood ratio. For instance, the loss

---

[10]This observation was made in the context of hypothesis testing in Ref. [24] and in the context of reweighting in Ref. [27–33]. It is likely that earlier references exist outside of HEP.

$$\text{Loss(NN)} = -\sum_{i \in S+B} \text{NN}(Y_i) + \frac{1}{2}\sum_{i \in B}\text{NN}(Y_i)^2, \tag{3.11}$$

has the property that NN $\propto \lambda(Y)$. The loss function proposed in Ref. [34] approaches $\log(\lambda(Y))$ directly, which may be useful when considering the logarithm of Eq. (3.8) for the statistical test. One potential advantage of learning the ratio first and then taking the logarithm is that one only needs to achieve proportionality while proportionality constants for the loss designed to learn the logarithm of $\lambda$ are suboptimal. See Appendix A for the derivations involving these loss functions.

Figure 3 illustrates the above concepts using the simple example from Sec. 2. These plots are trained only with $y = y_0 + y_1$ for simplicity. The left plot of Fig. 3 presents a histogram of the ratio of neural network outputs $f(y) = \text{NN}_1(y)/\text{NN}_2(y)$ for $\sigma = 0.08$. The neural network is parameterized in Keras [35] with the Tensorflow [36] backend with three fully connected hidden layers using 10, 20, 50 hidden nodes and the exponential linear unit activation function [37] with 10% dropout [38]. The last hidden layer is connected to a one-node output using the sigmoid activation function and the loss was binary cross-entropy. The networks are optimized using Adam [39] over three epochs with 500,000 examples and a batch size of 50. None of these parameters were optimized, as the problem is sufficiently simple that the specifics of the training are not important for the message presented with the results below.

As desired, the ratio of signal to background in Fig. 3 is monotonically increasing from left to right and this ratio is the same as the value on the horizontal axis. Since this problem is one-dimensional, it is possible to readily visualize the functional form of $f(y)$, shown in the right plot of Fig. 3. With a uniform background, the likelihood ratio should be simply the signal probability distribution, which is a Gaussian[11]. For comparison to the neural network, a binned version of the likelihood ratio is presented alongside the analytic result assuming $\sigma \ll 1$ so that edge effects are not relevant. The neural network output can be used to well-approximate the likelihood ratio.

Note that Eq. (3.9) is set up such that the classifier learns to separate $S+B$ from $B$. It is more common and often more pragmatic to train a classifier to distinguish $S$ from $B$ directly. The resulting classifier will be monotonically related to the one resulting from the $S+B$ versus $B$ classification [40]. Writing $p_{S+B}(Y) = \alpha p_S(Y) + (1-\alpha)p_B(Y)$ with $\alpha = S/(S+B)$, a neural network trained with the binary cross entropy will produce

$$\frac{\text{NN}_{S \text{ vs. } B}(Y)}{1-\text{NN}_{S \text{ vs. } B}(Y)} \propto \frac{p_S(Y)}{p_B(Y)} \tag{3.12}$$

$$= \frac{p_{S+B}(Y)-(1-\alpha)p_B(Y)}{\alpha p_B(Y)} \tag{3.13}$$

$$= \frac{1}{\alpha}\lambda(Y)-(1-\alpha). \tag{3.14}$$

Therefore, one can use the predictions for the yields $S$ and $B$ to correct the $S$ versus $B$ classifier.

A complete illustration of cuts-versus-bins-versus-deep learning is presented in Fig. 4 for the simple example from Sec. 2. As the above example shows that a NN can be used to well-approximate the likelihood ratio, the analytic result is used for this comparison. The horizontal

---

[11]In fact, this (train against a uniform distribution) can be used as a general method for using a neural network to learn a generic probability density. However, it becomes inefficient when the dimensionality of $y$ is large. One may be able to overcome this by using some localized (but known) density first before applying this procedure. Thank you to David Shih for useful discussions on this point.

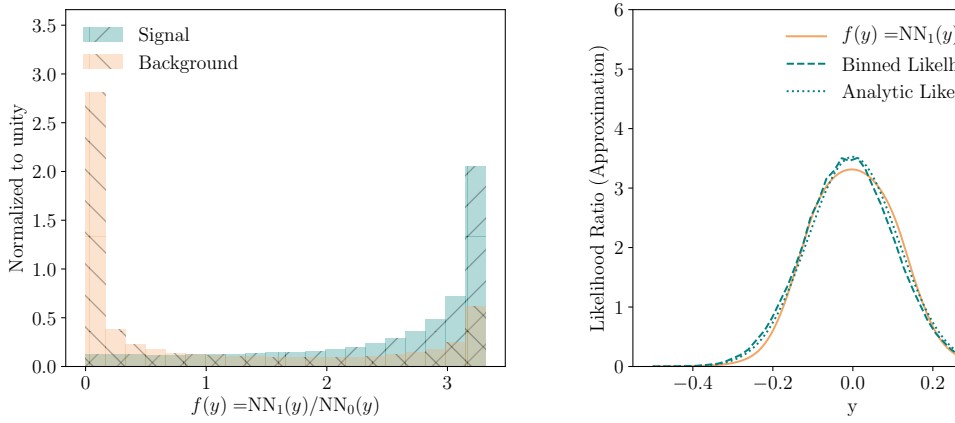

Figure 3: Results for a training with a fixed value of $\sigma = 0.08$. Left: histograms of the neural network outputs $f(y) = \mathrm{NN}_1(y)/\mathrm{NN}_2(y)$. Right: the functional form of $f(y)$ alongside a binned version of the likelihood ratio is presented alongside a Gaussian with mean zero and standard deviation 0.08.

axis in Fig. 4 is the 'level' or type I error of the test while the vertical axis is the power or (1-type II error rate). For the `Inclusive` scenario, $Y$ is not used and the test is based on $\lambda_{\mathrm{LR}}$ alone. For the other cases, the bins and cuts are based on the likelihood ratio. For the `Fixed cut`, the value is 0.5, and for the `Two bins` case, the bins boundaries are at 0.5 and 2, which were optimized to capture most of the information. The `Many bins` case uses 20 bins evenly spaced between 0 and 3. The `Optimal` procedure uses Eq. (3.8). For this simple example, two bins are nearly sufficient to capture all of the available information and by twenty bins, the procedure has converged to the one from Eq. (3.8).

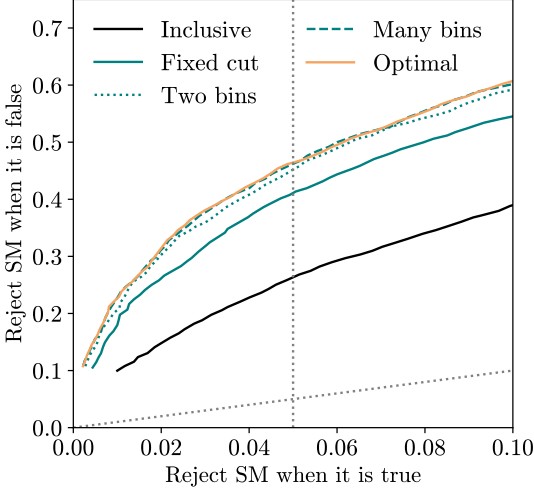

Figure 4: A comparison of different levels of information used to perform statistical tests for the $S+B$ hypothesis versus the background-only hypothesis. For illustration, a type I error of 0.05 is shown as a vertical dotted line and another line with random guessing at power = type I error is also shown as a dotted line. All of the $p$-values are computed with $10^4$ pseudo-experiments.

As a final observation, note that the proportionality of $\tilde{\lambda}(Y)$ with the likelihood ratio is strictly only true asymptotically when the NN is sufficiently flexible, there are enough training examples, etc. While modern deep learning models can often achieve a close approximation to $\tilde{\lambda}(Y)$ (see e.g. Ref. [33] for a high-dimensional example), further discussion about deviations in the optimality of the NN can be found in Sec. 4. An alternative (or complement) to engineering the NN output to be proportional to the likelihood ratio is to 'calibrate' the NN output in which the NN is viewed as an information-preserving dimensionality reduction and the class likelihood can be estimated numerically using one-dimensional density estimation methods (such as histogramming) [24]. An extensive guide to likelihood estimation using deep learning can additionally be found in Ref. [41–43].

### 3.3 Nuisance features

Given the high-dimensionality of LHC data, it is often necessary to only consider a redacted set of features for deep learning. It is tempting to only use features $Y$ for which $p(Y|S)/p(Y|B)$ is very different from unity. However, there are often features that are directly related to the resolution or uncertainty of other observables. Such features may have $p(Y|S)/p(Y|B) \approx 1$ on their own, but can enhance the potential of other observables when combined [44]. Examples of this type were mentioned in Sec. 2. A neural network approximation to the likelihood will naturally make the optimal use of these 'nuisance features'. Removing these features from consideration or even purposefully reducing their impact on the directly discriminative features [45] will necessarily reduce the analysis optimality. Nuisance features are not the same as nuisance parameters, where the value is unknown and a direct source of uncertainty. The interplay of nuisance parameters and neural network uncertainty is discussed in Sec. 4.3.

The simplest way to incorporate nuisance features is to simply treat them in the same way as directly discriminative features. Figure 5 illustrates this difference with the toy model from Sec. 2, where inference is performed with a classifier using only $Y_0 + Y_1$ and one using $Y_0 + Y_1$ and $\sigma$, where $\sigma$ is uniformly distributed between 0 and 0.29. As expected, when trying to determine $\mu$, the example that used $\sigma$ in addition to $Y$ is able to achieve a superior statistical precision. This case is nearly the same to the one where there is a global nuisance parameter $\mu$ and it is well constrained by some auxiliary data, i.e. constraining $\sigma$ with $Y_0 - Y_1$. In that case, one may use the techniques of parameterized classifiers [24, 25] to construct the NN, which is the same as treating $\sigma$ as a discriminating feature, only that a small number of $\sigma$ values may be available for training. This is discussed in more detail in Sec. 4.3.

## 4 Sources of Uncertainty

### 4.1 Overview

Uncertainty quantification is an essential part of incorporating deep learning into a HEP analysis framework. One of the most often-expressed phrases when someone proposes to use a deep neural network in an analysis is 'what is the uncertainty on that procedure?' The goal of this section is to be explicit about sources of uncertainty, how they impact the scientific result, and what can be done to reduce them.

There are generically two sources of uncertainty. One source of uncertainty decreases with more events (statistical or aleatoric uncertainty) and one represents potential sources of model bias that are independent of the number of events (systematic or epistemic uncertainty). These uncertainties are relevant for data as well as the models used to interpret the data, and in general there can be sources of uncertainty that have components due to both types.

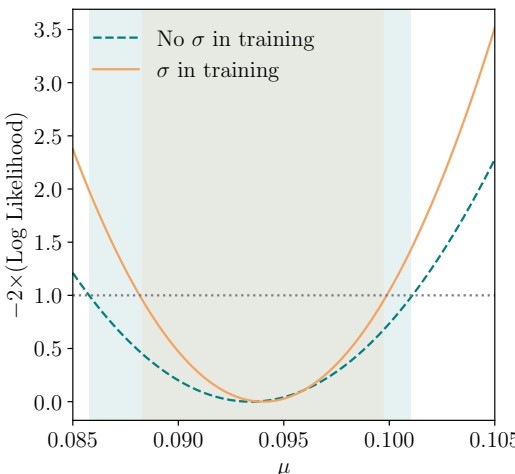

Figure 5: A determination of the signal strength $\mu$ using a classifier trained with or without $\sigma$, where $\sigma$ is uniformly distributed between 0 and 0.29. The true value of $\mu$ is 0.1 and $|\mathbf{Y}| = 10^4$. The shaded regions indicate the ranges where the curves intersect unity (the 68% confidence interval).

For most searches, the analysis strategy is designed prior[12] to any statistical tests on data ('unblinding'). In the deep learning context, this means that the neural network training is separate from the statistical analysis. As such, it is useful to further divide uncertainty sources into two more types: uncertainty on the precision/optimality of the procedure and uncertainty on the accuracy/bias of the procedure. These will be described in more detail below.

Consider the neural network setup from Sec. 3. If the network architecture is not flexible enough, there were not enough training examples, or the network was not trained for long enough, it may be that the likelihood ratio is not well-approximated. This means that the procedure will be **suboptimal** and will not achieve the best possible **precision**. However, if the classifier is well-modeled by the simulation, then $p$-values computed from the classifier may be **accurate**, which means that the results are **unbiased**. Conversely, a well-trained network may result in a biased result if the simulation used to estimate the $p$-value is not accurate. From the point of view of accuracy, the neural network is just a fixed non-linear high-dimensional function whose probability distribution must be modeled to compute $p$-values. In other words, the NN itself has no uncertainty in its accuracy - its evaluation is only uncertain through its inputs. A useful analogy is to consider common high-dimensional non-linear functions like the jet mass, which clearly have no uncertainty on their definition.

Figure 6 summarizes the various sources of uncertainty related to neural networks, broken down into the four categories described above. A machine learning model $NN(x)$ is trained on (usually) simulation following the distribution $p_{\text{train}}(x)$. Given the trained model, the probability density of $NN(x)$ is determined with another simulation following the distribution $p_{\text{prediction}}(x)$. It is often the case that $p_{\text{train}} = p_{\text{prediction}}$. Systematic uncertainties affecting the accuracy of the result originate from differences between $p_{\text{prediction}}$ and the true density $p_{\text{true}}$ while systematic uncertainties related to the optimality of the procedure originate from differences between $p_{\text{train}}$ and $p_{\text{true}}$.

The **precision/optimality** uncertainty is practically important for analysis optimization. If this uncertainty is large, one may want to modify some aspect of the analysis design (more

---

[12]This is not the case for a recent anomaly detection proposal where the event selection depends on the data [34, 46–55].

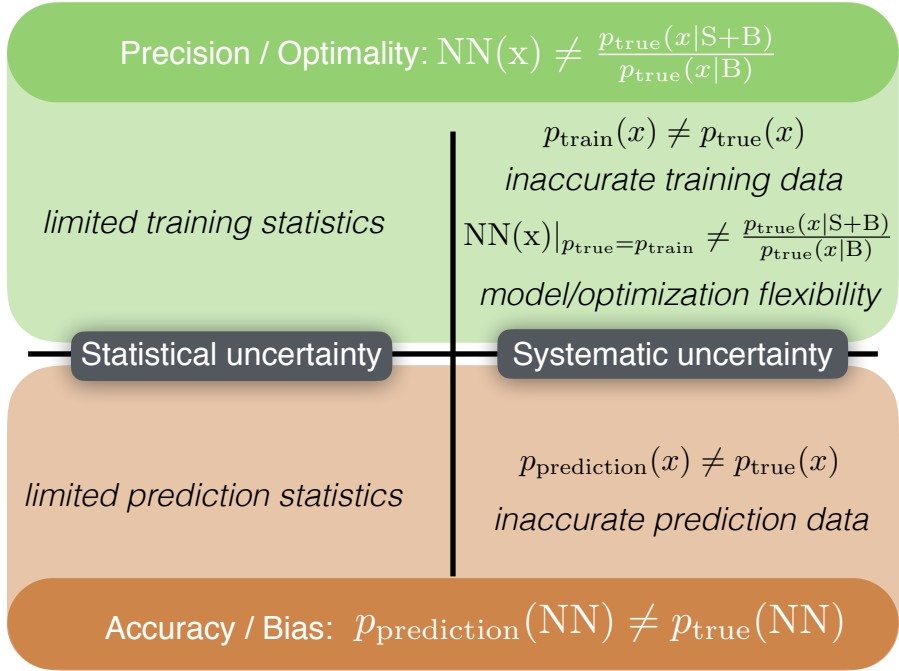

Figure 6: Sources of uncertainty affiliated with a neural network-based analysis. The symbol $p$ is used to represent a probability density function.

on this in Sec. 4.3). The precision/optimality uncertainty is often estimated by rerunning the training with different random initializations of the network parameters. This procedure is sensitive to both the finite size of the training dataset as well as the flexibility of the optimization procedure. One can also bootstrap the training data for fixed weight initialization to uniquely probe the statistical uncertainty from the training set size. An automated approach to estimate these uncertainties that does not require retraining multiple networks is Bayesian Neural Networks [56–60]. Estimating the uncertainty from the input feature accuracy can be performed by varying the inputs within their systematic uncertainty (see Sec. 4.4). This can be incorporated into network training via parameterized networks [24,25] with profiling (see Sec. 4.3). Determining the uncertainty from the model flexibility is challenging and there is currently no automated way for including this in the training. One (likely insufficient) possibility is to probe the sensitivity of the network performance to small perturbations in the network architecture.

Unless asymptotic formulae are used to directly estimate $p$-values with $\tilde{\lambda}$ (see Sec. 4.2), the optimality uncertainty is irrelevant from the perspective of scientific accuracy. To estimate the **accuracy/bias** uncertainty, the network is fixed and the test set inputs are varied. The statistical uncertainty can be estimated via bootstrapping [61]. Systematic uncertainties on the output are determined by varying (or profiling) the inputs within their individual uncertainties. As the whole point of deep learning is to exploit (possibly subtle) correlations in high dimensions, it is important to include the full systematic uncertainty covariance over the input feature space. This full matrix is often not known and impractically large, though parts of it can be factorized (see also Sec. 4.4).

Before turning to more specific details in the following sections, it is useful to consider efforts by the non-HEP machine learning community for uncertainties related to deep learning. An often-cited discussion of model uncertainty (not necessarily for deep learning) is Ref. [62], which lists seven sources of uncertainty. Many of these align well with those presented in Fig. 6. However, one key difference between HEP and industrial (and other scientific) applications

of deep learning is the high-quality of HEP simulation. For instance, consider a charged particle with momentum $p$ that is measured with momentum $p + \delta p$. Industrial applications may treat $\delta p$ as a source of uncertainty while in HEP, if $\delta p$ is well-modeled by the simulation, there is no uncertainty at all. Therefore, the tools and strategies for uncertainty in the machine learning literature are not always directly applicable to HEP. See e.g. Ref. [63] (and the many references therein) for a recent discussion of uncertainties related to deep learning models.

## 4.2 Asymptotic formulae with classifiers

Powerful results from statistics (e.g. Wilks' Theorem [64]) have made the use of asymptotic formulae for computing $p$-values widespread [65]. One could apply such formulae directly to Eq. (3.8) instead of estimating the distribution of the test statistic with pseudo-experiments with techniques like bootstrapping. This would require that the neural network learns *exactly* the likelihood ratio. Deviations of $\tilde{\lambda}(Y)$ from $\lambda(Y)$ will result in biased $p$-value calculations. In this case, it may be appropriate to combine (part of) the precision/optimality uncertainty with the accuracy/bias uncertainty in order to reflect the total uncertainty in the resulting $p$-value. However, this uncertainty on the $p$-value is completely reducible independent of the size of the precision/optimality uncertainty by using pseudo-experiments instead of asymptotic formulae so if this uncertainty is large, it is advised to simply[13] switch to pseudo-experiments.

## 4.3 Learning to profile: reducing the optimality systematic uncertainty

If the classification is particularly sensitive to a source of systematic uncertainty, one may want to reduce the dependence of the neural network on the corresponding nuisance parameter[14] - see e.g. Ref. [24] for an automated method for achieving this goal. While removing the dependence on such features may reduce model complexity, it will generally not improve the overall analysis sensitivity. By construction, if the classification is sensitive to a given nuisance parameter, removing the dependence on that parameter will reduce the nominal model performance. The significance will only degrade if the uncertainty is sufficiently large. To see this, suppose that there are two independent features to be used in training and one of them has an uncertainty for the background. In the asymptotic limit (including $S + B \gg 1$, $S \ll B$ [65]), the question of deriving additional benefit from the uncertain feature is given symbolically by

$$\frac{\epsilon_{S_1}\epsilon_{S_2}S}{\sqrt{\epsilon_{B_1}\epsilon_{B_2}B + (\delta_{\epsilon_{B_1}}\epsilon_{B_2}B)^2}} \stackrel{?}{>} \frac{\epsilon_{S_2}S}{\sqrt{\epsilon_{B_2}B}}, \tag{4.1}$$

where $\epsilon_{C_i}$ is the efficiency of classifier $i$ for class $C$ and $\delta_\epsilon$ is the uncertainty on the efficiency. Equation (4.1) is equivalent to

$$\frac{\epsilon_{S_1}}{\sqrt{\epsilon_{B_1}}}\left(1 - \frac{1}{2}\epsilon_{B_1}\epsilon_{B_2}\left(\frac{\delta_{\epsilon_{B_1}}}{\epsilon_{B_1}}\right)^2 B\right) \gtrsim 1. \tag{4.2}$$

Independent of the uncertainty, it only makes sense to use the classifier if the first term $\frac{\epsilon_{S_1}}{\sqrt{\epsilon_{B_1}}} > 1$. For the second term, if each of the efficiencies and relative uncertainties are $\mathcal{O}(10\%)$, then $B$ would need to be $\mathcal{O}(10^4)$ in order for the additional uncertain feature to detract from the analysis sensitivity. Even if the uncertainty is large, there are methods which construct classifiers

---

[13]This may require extensive computing resources, but given the growing availability of high performance computers with and without GPUs, hopefully this will become less of a barrier in the future.

[14]This is in contrast to 'nuisance features', where it is argued in Sec. 3.3 should never be removed from the neural network training except to reduce model complexity.

using the information about how they will be used ('inference-aware') and therefore should never do worse than the case where the uncertain features are removed from the start [66].

Especially for deep learning, one should proceed with caution when removing the dependence on single nuisance parameters that represent many sources of uncertainty. For example, it is common to use a single nuisance parameter to encode all of the fragmentation uncertainty. This is already tenuous when using high-dimensional, low-level inputs, as the uncertainty covariance is highly constrained. If the sensitivity to such a nuisance parameter is removed, it does not mean that the network is insensitive to fragmentation - it only means that it is not sensitive to the fragmentation variations encoded by the single nuisance parameter. This may also apply to other sources of theory uncertainty such as scale variations for estimating uncertainties from higher-order effects. In some cases, higher-order terms may be known to be small and can justify reducing the sensitivity to scale variations [67], but these terms are typically not known.

The above arguments can be complicated when the two features are not independent and the background is estimated entirely from data via the ABCD method[15] or a sideband fit. In that case, the strength of the feature dependence can increase the background uncertainty. When the dependence is strong enough, it may no longer be possible to estimate the background. There are a variety of neural network [68,69] and other [70–73] approaches to achieve this decorrelation.

Instead of removing the dependence on uncertain features, a potentially more powerful way to reduce precision systematic uncertainties is to do exactly the opposite - depend explicitly on the nuisance parameters [24,25]. By parameterizing a neural network $f_\theta$ as a function of the nuisance parameters $\theta$, one can achieve the best performance for each value of $\theta$ (such as the $\pm 1\sigma$ variations). Furthermore, this can be combined with profiling so that when the data are fit to determine $\theta$ and constrain its uncertainty, the neural network is accordingly modified. The left plot of Fig 7 shows that the idea of parameterized classifiers [24,25] works well for the toy example from Sec. 2. The training was performed with values $\sigma = 0.02, 0.04, 0.08, 0.16, 0.32$. The right plot of Fig. 7 shows the uncertainty on $\mu$ when performing a statistical test with a neural network trained with a sample generated with nuisance parameter $\sigma'$. As expected, the uncertainty is smallest when $\sigma' = \sigma$ so that $f_\sigma$ is the optimal classifier for that value of the nuisance parameter (clearly, the uncertainty is worse when $\sigma$ is large). Therefore, if the fitted value of $\sigma$ is the true value (as is hopefully true when it is profiled), the statistical procedure will make the best use of the data.

In practice, it may be challenging to generate multiple training datasets with different values of $\sigma$. Neural networks are excellent at interpolating between parameter values, but there must be enough $\sigma$ values to ensure an accurate interpolation. This can be especially challenging if $\sigma$ is multi-dimensional. In practice, learning to profile will likely work well for nuisance parameters that only require a variation in the final analysis inputs (such as the jet energy scale variation) and not for parameters that require rerunning an entire detector simulation (such varying fragmentation model parameters). For the latter case, one may be able to use high-dimensional reweighting to emulate parameter variations without expensive detector simulations [33].

## 4.4 High-dimensional bias uncertainties

The single biggest challenge to using high-dimensional features for neural networks is estimating high-dimensional uncertainties. Many sources of experimental uncertainty factorize into independent terms for each object. However, physics modeling uncertainties are often

---

[15]For this method, there are two independent features which are used to form four regions called $A, B, C, D$ based on threshold cuts on the two features. One of these regions is enriched in signal and the other three are used to estimate the background.

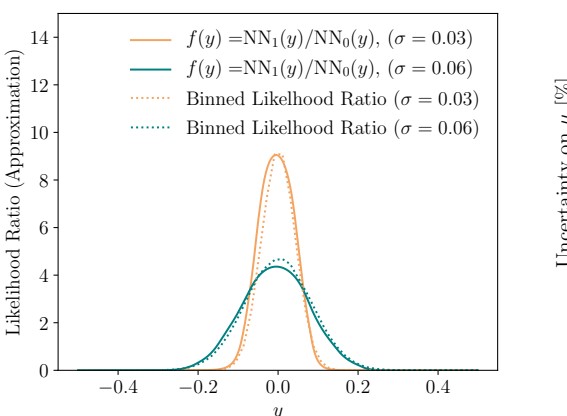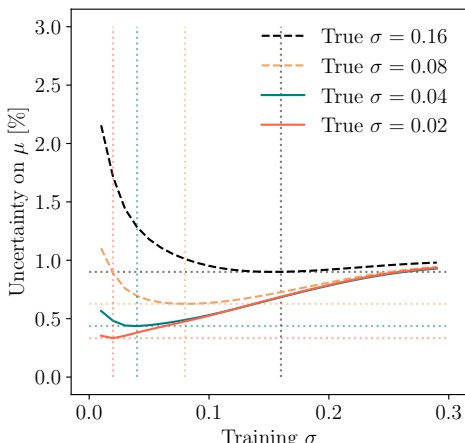

Figure 7: Left: The functional form of $f_\sigma(y)$ alongside a binned version of the likelihood ratio for values of $\sigma$ not presented to the neural network during training: $\sigma \in \{0.03, 0.06\}$. Right: The statistical uncertainty on the value of $\mu$ when performing a statistical test with $f_{\sigma'}(y)$ where $\sigma'$ is the value of the nuisance parameter used in the training. The true value of $\sigma$ (the one used in testing) is indicated by vertical dashed lines. This test used $|\mathbf{Y}| = 10^4$.

grouped into two-point variations that cover many physical effects all at once. These uncertainties may no longer be appropriate when the input features are high-dimensional (see also Sec. 4.3). There are additional complications when computing uncertainties beyond $1\sigma$ and even for the $1\sigma$ uncertainties if the NN is a non-monotonic transformation of the input as quantiles are not preserved.

The fact that this section is short is an indication that new ideas are needed in this area.

## 5 Conclusions and Outlook

This paper has reviewed how deep learning can be used to make the best use of data for new particle searches at the LHC. Deep learning-based classifiers can serve as surrogates to the likelihood ratio in order to achieve an optimal test statistic. Nuisance features can improve the performance of such classifiers even if they are not individually useful for distinguishing signal and background. The ways in which uncertainties affects deep learning-based inference were discussed and categorized into *precision uncertainties* related to the optimality of the procedure and *accuracy uncertainties* related to the bias of the method. While both sources of uncertainty are useful to quantify, the latter is much more important for the utility of the results. Precision uncertainties can be reduced by letting the deep learning models depend explicitly on the nuisance parameters and then profiling them during the statistical analysis.

As deep learning-based search strategies become more common, it will be important to discuss all of these topics in more detail and develop strategies to ensure that the precious data from the LHC are used in the best way possible to learn the most about the fundamental properties of nature.

## Acknowledgments

BPN would like to thank A. Andreassen J. Collins, K. Cranmer, D. Finkbeiner, M. Kagan, E. Metodiev, L. de Oliveira, M. Paganini, T. Plehn, M. Spannowsky, D. Shih, F. Tanedo, and J. Thaler, and D. Whiteson, for stimulating conversations, D. Whiteson, D. Shih, K. Cranmer, and T. Plehn for detailed comments on the manuscript, and D. Whiteson and F. Tanedo, for their hospitality at UC Irvine and Riverside, respectively. This work was supported by the U.S. Department of Energy, Office of Science under contract DE-AC02-05CH11231 and by a fundamental physics innovation award from the Gordon and Betty Moore Foundation. This work was completed at the Aspen Center for Physics, which is supported by National Science Foundation grant PHY-1607611.

## A  Loss functions and asymptotic behavior

The results presented here[16] can be found (as exercises) in textbooks, but are repeated here for completeness. Let $X$ be some discriminating features and $Y \in \{0, 1\}$ is another random variable representing class membership (signal versus background). Consider the general problem of minimizing an average loss for the function $f(x)$:

$$f = \operatorname{argmin}_{f'} \mathbb{E}[\operatorname{loss}(f'(X), Y)], \tag{A.1}$$

where $\mathbb{E}$ means 'expected value', i.e. average value or mean (sometimes represented as $\langle \cdot \rangle$). The expectation values are performed over the joint probability density of $(X, Y)$. One can rewrite Eq. (A.1) as

$$f = \operatorname{argmin}_{f'} \mathbb{E}[\mathbb{E}[\operatorname{loss}(f'(X), Y)|X]. \tag{A.2}$$

The advantage[17] of writing the loss as in Eq. (A.2) is that one can see that it is sufficient to minimize the function (and not functional) $\mathbb{E}[\operatorname{loss}(f'(x), Y)|X = x]$ for all $x$. To see this, let $g(x) = \operatorname{argmin}_{f'} \mathbb{E}[\operatorname{loss}(f'(x), Y)|X = x]$ and suppose that $h(x)$ is a function with a strictly smaller loss in Eq. (A.2) than $g$. Since the average loss for $h$ is below that of $g$, by the intermediate value theorem, there must be an $x$ for which the average loss for $h$ is below that of $g$, contradicting the construction of $g$.

As a first concrete example, consider the mean-squared error loss: $\operatorname{loss}(f'(X), Y) = (f'(X) - Y)^2$. One can compute

$$g(x) = \operatorname{argmin}_{f'} \mathbb{E}[\operatorname{loss}(f'(x), Y)|X = x] \tag{A.3}$$

$$= \operatorname{argmin}_{f'} \mathbb{E}[(f'(x) - Y)^2|X = x] \tag{A.4}$$

$$= \operatorname{argmin}_{f'} \mathbb{E}[(f'(x))^2 + Y^2 - 2f'(x)Y|X = x] \tag{A.5}$$

$$= \operatorname{argmin}_{f'} \left((f'(x))^2 + \mathbb{E}[Y^2|X = x] - 2f'(x)\mathbb{E}[Y|X = x]\right) \tag{A.6}$$

$$= \operatorname{argmin}_{f'} \left((f'(x))^2 - 2f'(x)\mathbb{E}[Y|X = x]\right) \tag{A.7}$$

$$= \operatorname{argmin}_{z} \left(z^2 - 2z\mathbb{E}[Y|X = x]\right), \tag{A.8}$$

---

[16]Heavily borrowed from the appendix in Ref. [33].

[17]The derivation below for the mean-squared error was partially inspired by Appendix A in Ref. [24].

where the last line follows since $f'(x)$ is simply a number. The value of $z$ that minimizes $z^2 - 2z\mathbb{E}[Y|X = x]$ is simply $\mathbb{E}[Y|X = x]$, leading to the well-known result that the mean-squared error results in the average value of the target[18]. Since $Y$ is binary $\mathbb{E}[Y|X = x] = p(Y = 1|X)$, the conditional probability. Similarly for binary cross-entropy:

$$g(x) = -\text{argmin}_{f'}\mathbb{E}\left[Y\log\big(f'(x)\big) + (1-Y)\log\big(1 - f'(x)\big)|X = x\right] \tag{A.9}$$

$$= -\text{argmin}_{f'}\big(\mathbb{E}[Y|X = x]\log\big(f'(x)\big) + (1 - \mathbb{E}[Y|X = x])\log\big(1 - f'(x)\big)\big) \tag{A.10}$$

$$= -\text{argmin}_{z}\big(\mathbb{E}[Y|X = x]\log(z) + (1 - \mathbb{E}[Y|X = x])\log(1 - z)\big). \tag{A.11}$$

The derivative of the last line is

$$\frac{\mathbb{E}[Y|X = x]}{z} - \frac{1 - \mathbb{E}[Y|X = x]}{1 - z} = 0 \implies z = \mathbb{E}[Y|X = x], \tag{A.12}$$

where again, the optimal value is $p(Y = 1|X)$. The same analysis can be applied to the loss in Eq. (3.11):

$$g(x) = -\text{argmin}_{f'}\mathbb{E}\left[Yf'(x) - \frac{1}{2}(1 - Y)f'(x)^2\bigg|X = x\right] \tag{A.13}$$

$$= -\text{argmin}_{f'}\left(\mathbb{E}[Y|X = x]z - \frac{1}{2}(1 - \mathbb{E}[Y|X = x])z^2\right). \tag{A.14}$$

The derivative of the last line is

$$\mathbb{E}[Y|X = x] - (1 - \mathbb{E}[Y|X = x])z = 0 \implies z = \frac{\mathbb{E}[Y|X = x]}{1 - \mathbb{E}[Y|X = x]}. \tag{A.15}$$

Equation (3.10) in the text shows that the above is proportional to the likelihood ratio.

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
