# Peer review of "A guide for deploying Deep Learning in LHC searches: How to achieve optimality and account for uncertainty"

_SciPost Physics, doi:SciPost Phys. 8, 090 (2020)_

## Round 1 · Referee Report · Anonymous (Referee 1) · 2019-11-5

Strengths

1-Addresses relevant systematics related to the application of neural networks to (high energy) physics data analysis and hypothesis testing.

2-Provides simple yet clear examples for all the concepts introduced and uses these to illustrate and compare traditional and supervised machine learning approaches to likelihood estimation.

Weaknesses

1-The main concepts of "Deep Learning" and "modern machine learning" are not clearly and unambiguously defined. In the paper they are used synonymously with neural networks used for likelihood approximation, which is a much more narrower concept. A more accurate approach would be to consistently use the term "neural networks" instead throughout the paper, including the title.

2-In the discussion on the uncertainties of the neural network inputs on page 12, it is claimed that if these are well modeled by the Monte Carlo simulation, there is no residual source of systematic uncertainty when such inputs are used to train a neural network likelihood approximator. However, in practice there is always some degree of mismodeling of the input uncertainty distributions. From the existing discussion it is not clear what impact such mismodelling can have on the neural network outputs, how to estimate, and potentially mitigate it. For comparison in the case of a simple likelihod ratio, the impact of an (over/under)estimated input uncertainty is transparent and can be evaluated using e.g. nuisance parameter estimation.

Report

In the submitted manuscript the author addresses relevant systematics related to the application of neural networks to (high energy) physics data analysis and hypothesis testing - in particular related to optimality and systematic uncertainty. Using simple yet clear examples for all the concepts introduced, the paper illustrates and compares traditional and supervised machine learning approaches to likelihood estimation in high energy physics highlighting potential sources of sub-optimality and systematic bias.

The paper is timely and very relevant in light of the recent advancement and proliferation of machine learning approaches to likelihood estimation in high energy physics.

However, the paper could be improved in terms of clarity and conciseness. In particular, the main concepts of "Deep Learning" and "modern machine learning" are not clearly and unambiguously defined. In the paper they are used synonymously with neural networks used for likelihood approximation, which is a much more narrower concept. A more accurate approach would be to consistently use the term "neural networks" instead throughout the paper, including the title.

Perhaps a more important weakness of the present discussion is related to the uncertainties of the neural network inputs. On page 12 it is claimed that if these are well modeled by the Monte Carlo simulation, there is no residual source of systematic uncertainty when such inputs are used to train a neural network likelihood approximator. However, in practice there is always some degree of mismodeling of the input uncertainty distributions. From the existing discussion it is not clear what impact such mismodelling can have on the neural network outputs, how to estimate, and potentially mitigate it. For comparison in the case of a simple likelihod ratio, the impact of an (over/under)estimated input uncertainty is transparent and can be evaluated using e.g. nuisance parameter estimation.

Finally, there are several minor issues with the referencing and definitions as follows:

-Reference [13] lacks scientific rigor. The author should at least provide a general reference to a recent review or a set of foundational papers.

-$CL_{S+B}$ and $CL_{B}$ below Eq. (3.2) on page 4 are not defined.

-In the sentence starting on line 2 of page 13 with "One could apply...", the term "toys" is not defined.

Requested changes

1-The main concepts of "Deep Learning", "modern machine learning" should be clearly and concisely defined within the narrower scope used in the present paper, i.e. neural networks as likelihood approximators, addressing the point 1-weakness.

2-Reference [13] lacks scientific rigor. The author should at least provide a general reference to a recent review or a set of foundational papers.

3-$CL_{S+B}$ and $CL_{B}$ below Eq. (3.2) on page 4 are not defined.

4-The discussion on the uncertainties of the neural network inputs and their impact on the likelihood estimation should be extended, addressing the point 2-weakness.

5-In the sentence starting on line 2 of page 13 with "One could apply...", the term "toys" is not defined.

  • validity: good
  • significance: high
  • originality: good
  • clarity: high
  • formatting: good
  • grammar: excellent

Author:  Benjamin Nachman  on 2020-03-10  [id 759]

(in reply to Report 1 on 2019-11-05)

Dear Referee,

Thank you for taking the time to review my manuscript! I am happy to hear that you find the paper is timely and relevant. I appreciate your feedback and I have produced a new version of the manuscript that incorporates your requested changes. Below is a detailed response, with my answers offset by a vertical line. Thank you again for your time.

Sincerely, Ben

The main concepts of "Deep Learning", "modern machine learning" should be clearly and concisely defined within the narrower scope used in the present paper, i.e. neural networks as likelihood approximators, addressing the point 1-weakness.

Footnote 1 has now been clarified: "Here, `deep learning' means machine learning with modern neural networks. These networks are deeper and more flexible than artificial neural networks from the past and can now readily process high-dimensional feature spaces with complex structure. In the context of classification, deep neural networks are powerful high-dimensional likelihood-ratio approximators, as described in later sections. Machine learning has a long history in HEP and there are too many references to cite here -- see for instance Ref. [1]".

Reference [13] lacks scientific rigor. The author should at least provide a general reference to a recent review or a set of foundational papers.

This has been replaced with a reference to recent reviews and list of papers that covers many directions of the deep learning for HEP literature.

CL_{S+B} and CL_{B} below Eq. (3.2) on page 4 are not defined.

These are now explicitly defined with their own equations.

The discussion on the uncertainties of the neural network inputs and their impact on the likelihood estimation should be extended, addressing the point 2-weakness.

Point 2 under weaknesses: In the discussion on the uncertainties of the neural network inputs on page 12, it is claimed that if these are well modeled by the Monte Carlo simulation, there is no residual source of systematic uncertainty when such inputs are used to train a neural network likelihood approximator. However, in practice there is always some degree of mismodeling of the input uncertainty distributions. From the existing discussion it is not clear what impact such mismodelling can have on the neural network outputs, how to estimate, and potentially mitigate it. For comparison in the case of a simple likelihod ratio, the impact of an (over/under)estimated input uncertainty is transparent and can be evaluated using e.g. nuisance parameter estimation.

Page 12 contrasts systematic uncertainties in HEP versus other scientific and industrial applications. I suspect you are instead concerned about the previous page and your comments seem to refer to what I'm calling "precision/optimality" uncertainties. p_train is not p_true, then the NN will be suboptimal and the analysis will not have the best possible sensitivity. This is not a problem from the point of view of accuracy (as I argue on p11), but is not the best use of the data. Sec. 4.3 is about profiling this uncertainty.

In the sentence starting on line 2 of page 13 with "One could apply...", the term "toys" is not defined.

This has been changed to "pseudo-experiments" and it is further clarified that these can be created with methods like bootstrapping.

---

## Round 1 · Referee Report · Anonymous (Referee 2) · 2020-1-10

Strengths

1- Several interesting points made on a very crucial topic 2- Addresses typical points raised in the community, every time one shows machine-learning-based solutions to an audience not familiar with machine learning 3- It could become a useful reference to many other papers

Weaknesses

1- Insufficient development of many of the points made, which would have required some more rigorous discussion with examples

2- the example discussed on Page 3 seems to me an oversimplification of the problem, when a more realistic one could be done very easily

  • why a flat background shape?
  • why a constant sigma?
  • why a counting experiment and not a bump hunt, for instance?

Report

The paper touches a very relevant issue, at a moment in which the HEP community is looking at Machine Learning as a production-ready tool for many problems. The problems of incorporating systematic uncertainties and how to reach optimal performance in presence of systematic effects are frequently raised and often confused. In this respect, this paper comes at a good point in time, since it offers a clarification on this specific point.

On the other hand, the current version of the paper is a mixture of original studies with toy examples and statements that one can clearly agree with but that are not substantiated with a fact-based discussion. Moreover, many of these arguments have been heard in technical workshops. The reader is left with fundamental questions on the nature of this paper. Is it a review? Or is its content to be intended as 100% original. I have the impression that the current version is a little bit a mixture of the two, which might not be the case. I would suggest the author to make an effort to clarify this aspect and highlight the novel nature of the material presented.

More comments and suggestions on specific parts of the paper are given below.

Requested changes

On page 1: Modern machine learning can be ambiguous.There is modern ml research that doesn’t use nns at all (e.g., some quantum version of pre-deep-learning algorithms). I would just say ‘deep learning’ to indicate the research on artificial neural networks post 2012.

On page 2: sigma_i = sigma seems more simple than the case of jet-energy scale resolution. I have the impression that with jets the resolution is not just a constant value (e.g., it might depend on jet pT)

On page 4: NP gives the most powerful test for simple hypotheses. Since the paper discusses optimal performance in presence of uncertainties, the hypotheses under consideration are never simple (i.e., there are nuisance parameters to take into account). This point should be stated.

On page 4: the discussion on CLs is very CMS/ATLAS-centric. In my recollection, the issue with CLs+b that motivated the use of CLs was the problem of empty sets and the breaking of coverage. This was a big issue in neutrino physics in the 90s (the issue of negative square of neutrino mass). Also, there are Bayesian methods as well that were used by other experiments. I would not restrict hep to common practice in atlas & cms

On page 5: the comment about models not being know analytically seems again very cms & atlas centric. There is a pre-LHC literature, as large as the LHC one, that used analytical models in unbinned ML fits as a standard.

Eq 3.7: isn't this assuming that one between H1 and H0 have to be right? What does that imply?

Footnote 8: this is an example of a statement that should be expanded with examples (see general comments). Otherwise it remains a puzzling statement which doesn't add anything concrete to the paper.

On page 11: In my personal opinion, the statement in the middle of the page ("From the point of view...") is the most convincing answer to many skeptics. It should be highlighted in a stronger way, because people confuse too often non-optimal models with wrong models. Maybe it should be moved to the introduction (where I would put a paragraph clarifying the main issue the paper is trying to address and the original content of the discussion that follows)

On page 13: I have the impression that "toys" is hep jargon

On references: 1) I think that the reviews refereed to in [4,-6] are as relevant as the papers they are based on. I would add "and references therein" or something like that. 2) I was surprised by Ref [13], which I find very unorthodox in a community that is centred around the concept of giving credit through citations, something that the author is usually very sensitive to. I think that the more effort should go on this point 3) ref. [1] is not the best choice for ML in HEP. There are papers that date back to 1989 and hundreds of papers written in the 2000s that used NNs (e.g., B factories) before TMVA was available. 4) I would move Ref 12 to a footnote

  • validity: high
  • significance: good
  • originality: ok
  • clarity: good
  • formatting: excellent
  • grammar: excellent

Author:  Benjamin Nachman  on 2020-03-10  [id 758]

(in reply to Report 3 on 2020-01-10)

Dear Referee,

Thank you for taking the time to review my manuscript! I am happy to hear that you find the paper to make several interesting points made on a very crucial topic and that it could become a useful reference to many other papers. I appreciate your feedback and I have produced a new version of the manuscript that incorporates your requested changes. Below is a detailed response, with my answers offset by a vertical line. Thank you again for your time.

Sincerely, Ben

Is it a review? Or is its content to be intended as 100\% original. I have the impression that the current version is a little bit a mixture of the two, which might not be the case. I would suggest the author to make an effort to clarify this aspect and highlight the novel nature of the material presented.

I tried to make this clear with the statement: "The exposition is based on a mixture of old and new insights..." in the introduction. I think that most of the key concepts are already present in the literature (although mostly scattered and often hidden). My impression is that Fig. 6 is a new way of thinking about uncertainties for NNs and I think this is also true for various places in the paper. None of the examples or prose are taken from another reference. I am not sure of another way to make this clear other than the current statement in the introduction, but I am happy to make a modification if you have a concerete suggestion.

On page 1: Modern machine learning can be ambiguous.There is modern ml research that doesn’t use nns at all (e.g., some quantum version of pre-deep-learning algorithms). I would just say ‘deep learning’ to indicate the research on artificial neural networks post 2012.

This has now been clarified, thank you for your comment.

On page 2: $\sigma_i = \sigma$ seems more simple than the case of jet-energy scale resolution. I have the impression that with jets the resolution is not just a constant value (e.g., it might depend on jet pT)

I have now clarified what I mean: "In practice, $\sigma$ may vary from event-to-event in a known way, but there is one global nuisance parameter."

On page 4: NP gives the most powerful test for simple hypotheses. Since the paper discusses optimal performance in presence of uncertainties, the hypotheses under consideration are never simple (i.e., there are nuisance parameters to take into account). This point should be stated.

This has been clarified.

On page 4: the discussion on CLs is very CMS/ATLAS-centric. In my recollection, the issue with CLs+b that motivated the use of CLs was the problem of empty sets and the breaking of coverage. This was a big issue in neutrino physics in the 90s (the issue of negative square of neutrino mass). Also, there are Bayesian methods as well that were used by other experiments. I would not restrict hep to common practice in atlas & cms

The motivation of CLs is that one does not want to exclude a model when CLb is small i.e. if the background model is bad, then probably background + signal is also bad, regardless of if there is no signal. It was first introduced at LEP. I do not use CLs anywhere else other than to mention it here and it is the community standard (used not only for ATLAS and CMS, but for a majority of searches in HEP across experiments). Hopefully it is okay to leave these sentences untouched.

On page 5: the comment about models not being know analytically seems again very cms & atlas centric. There is a pre-LHC literature, as large as the LHC one, that used analytical models in unbinned ML fits as a standard.

I have now clarified that I am thinking of Y as high-dimensional where we basically never have an analytic likelihood (instead, we have to use simulations). This is not ATLAS/CMS-centric, but a generic property of HEP experiments that have complex final states.

Eq 3.7: isn't this assuming that one between H1 and H0 have to be right? What does that imply?

That is correct - this is a simple hypothesis test. The reason for CLs is exactly to help when the alternative hypothesis is also not true. The derivation here is just to show that there is a connection between the binned and unbinned case, so if you think the likelihood ratio is a useful test statistic, then Eq. 3.7 should be motivated.

Footnote 8: this is an example of a statement that should be expanded with examples (see general comments). Otherwise it remains a puzzling statement which doesn't add anything concrete to the paper.

The paper is not about generative models, so I would prefer to keep this as a footnote as it is not relevant for the main discussion. For this case, Fig. 3 is actualy an example - you can see that the NN has learned the PDF by training a Gaussian versus a uniform distribution. If you think this footnote is too distracting, I could remove it.

On page 11: In my personal opinion, the statement in the middle of the page ("From the point of view...") is the most convincing answer to many skeptics. It should be highlighted in a stronger way, because people confuse too often non-optimal models with wrong models. Maybe it should be moved to the introduction (where I would put a paragraph clarifying the main issue the paper is trying to address and the original content of the discussion that follows)

I am very glad to hear this and thank you for the suggestion! I have added a couple of sentences at the end of the introduction as per your suggestion.

On page 13: I have the impression that "toys" is hep jargon

This has been changed to "pseudo-experiments" and it is further clarified that these can be created with methods like bootstrapping.

On references: 1) I think that the reviews refereed to in [4,-6] are as relevant as the papers they are based on. I would add "and references therein" or something like that.

Agreed and added as per your suggestion.

2) I was surprised by Ref [13], which I find very unorthodox in a community that is centred around the concept of giving credit through citations, something that the author is usually very sensitive to. I think that the more effort should go on this point

Thank you for this comment - I struggled with a compromise here and I apologize for going too far in one direction. This has been replaced with a reference to recent reviews and list of papers that covers many directions of the deep learning for HEP literature. I hope you find the new version to be more inclusive and representative of the great work that has been done in this community. I still apologize for not citing everyone.

3) ref. [1] is not the best choice for ML in HEP. There are papers that date back to 1989 and hundreds of papers written in the 2000s that used NNs (e.g., B factories) before TMVA was available.

I have modified this to make it clear that TMVA was not the first. Hopefully the new footnote is more inclusive.

4) I would move Ref 12 to a footnote

Now that the "pheno" reference is now much longer, the footnote would take up a lot of room so perhaps it is best to keep this and Ref. 13 as references.

---

## Round 1 · Referee Report · Anonymous (Referee 3) · 2020-1-13

Strengths

  • Symbolic derivations linking network output to the likelihood ratio
  • Using knowledge of the likelihood ratio to show how systematic and statistical uncertainties impact the scientific result when using neural networks.

Weaknesses

  • The different sums in equation 3.3 should be explained. The sum of i in patches P is easy, but the sum over j from 1 to n does not seem to follow from equation 3.1.
  • Potentially confusing notation, with different uses of $\sigma$ (cross section many times but also the width of the gaussian error for the illustrative model).

Report

This manuscript presents a categorization of uncertainties that can be encountered in high energy physics in the context of machine learning. Within this classification, ways to tackle the uncertainties are presented (if they exist) or there is a call for more studies (if they do not exist). There are many strengths of this manuscript, including Figure 6 and the numerous examples. The main takeaway is that a neural network's output can be mapped to the likelihood ratio, which can be used for the most powerful statistical tests.

Although there are many strengths, I find the overall presentation underwhelming. It is difficult to tell what audience the author is aiming for. The title suggests a guide, in which case I would expect more detail in the examples. Using the illustrative model, the author highlights features of how the neural network output can be used in Figures 4 and 5. For this to be a "guide", it would be beneficial to give more detail about the specific methodology and MC toys. Similarly, Figure 7 is presented as proof that the idea works but doesn't offer a guide on how to perform the test. Who is this a guide for? It was also odd to find a specific sentence that the neural network hyperparameters were not optimized in the section about achieving optimality; the author could draw a more precise definition of optimal as the likelihood versus the minimum of the loss function.

Some of the terms in equations are left unexplained, such as 3.3. The main test statistic, $\lambda_{\rm{LR}}$ is only given an inline definition, while the example of what the paper is not going to explore is given an equation number.

Requested changes

  • The likelihood ratio $\lambda_{\rm{LR}}$ is the chosen test statistic, but does not get its own equation number, while the profile likelihood ratio $\lambda_{\rm{PLR}}$ is not used, but does get an equation number. This should be changed, as it is confusing when looking back to see what is being used.
  • More detailed explanation of equation 3.3.

  • validity: good
  • significance: good
  • originality: good
  • clarity: ok
  • formatting: excellent
  • grammar: excellent

Author:  Benjamin Nachman  on 2020-03-10  [id 757]

(in reply to Report 2 on 2020-01-13)
Category:
answer to question

Dear Referee,

Thank you for taking the time to review my manuscript! I am happy to hear that you find Fig. 6 to be a strength of the paper as I find this to summarize the main message. I appreciate your feedback and I have produced a new version of the manuscript that incorporates your requested changes. Below is a detailed response, with my answers offset by a vertical line. Thank you again for your time.

Sincerely, Ben

The likelihood ratio $\lambda_\text{LR}$ is the chosen test statistic, but does not get its own equation number, while the profile likelihood ratio $\lambda_\text{PLR}$ is not used, but does get an equation number. This should be changed, as it is confusing when looking back to see what is being used.

Thank you for this suggestion - $\lambda_\text{LR}$ now has its own equation.

More detailed explanation of equation 3.3.

This has now been expanded.

---

## Round 3 · Referee Report · Anonymous (Referee 1) · 2020-5-29

Report

In his response and the updated version of the manuscript the author has successfully addressed all concerns raised in my first report.

---

## Round 3 · Referee Report · Anonymous (Referee 2) · 2020-5-31

Report

I reviewed the new version of the paper and I appreciate the effort made by the author to reply to my points raised. A few of them were general remarks (e.g., what was CLs introduced for, what is common practice in HEP within vs beyond the LHC community, now vs the past). While I still disagree on a few of these points (not sure if I made me clear, judging from the answer), I think that these are very minor aspects and I would not slow down the publication process for them.

The message of the paper is correct and it clearly meets the expectations of this journal, so I am happy to recommend its publication.

---

## Round 3 · List of Changes

I have updated the manuscript in response to the reviewer comments. A detailed response to each reviewer can be found on the reviewer reports page.

---

## Editorial Decision

published